# EEG Signal Processing and Supervised Machine Learning to Early Diagnose Alzheimer's Disease

Daniele Pirrone [1], Emanuel Weitschek [1], Primiano Di Paolo [1], Simona De Salvo [2] and Maria Cristina De Cola [2,*]

1   Department of Engineering, Uninettuno University, Corso Vittorio Emanuele II 139, 00186 Rome, Italy; daniele.pirrone@uninettunouniversity.net (D.P.); emanuel.weitschek@uninettunouniversity.net (E.W.); p.dipaolo@students.uninettunouniversity.net (P.D.P.)
2   IRCCS Centro Neurolesi "Bonino-Pulejo", Contrada Casazza, SS113, 98124 Messina, Italy; simona.desalvo@irccsme.it
*   Correspondence: mariacristina.decola@irccsme.it; Tel.: +39-090-60128141

**Abstract:** Electroencephalography (EEG) signal analysis is a fast, inexpensive, and accessible technique to detect the early stages of dementia, such as Mild Cognitive Impairment (MCI) and Alzheimer's disease (AD). In the last years, EEG signal analysis has become an important topic of research to extract suitable biomarkers to determine the subject's cognitive impairment. In this work, we propose a novel simple and efficient method able to extract features with a finite response filter (FIR) in the double time domain in order to discriminate among patients affected by AD, MCI, and healthy controls (HC). Notably, we compute the power intensity for each high- and low-frequency band, using their absolute differences to distinguish among the three classes of subjects by means of different supervised machine learning methods. We use EEG recordings from a cohort of 105 subjects (48 AD, 37 MCI, and 20 HC) referred for dementia to the IRCCS Centro Neurolesi "Bonino-Pulejo" of Messina, Italy. The findings show that this method reaches 97%, 95%, and 83% accuracy when considering binary classifications (HC vs. AD, HC vs. MCI, and MCI vs. AD) and an accuracy of 75% when dealing with the three classes (HC vs. AD vs. MCI). These results improve upon those obtained in previous studies and demonstrate the validity of our approach. Finally, the efficiency of the proposed method might allow its future development on embedded devices for low-cost real-time diagnosis.

**Keywords:** Alzheimer's disease; EEG signals; power spectrum; FIR filtering; supervised machine learning

## 1. Introduction

Alzheimer's diseases (AD) belong to the class of dementia, a neurodegenerative disease characterized by a range of impairments in brain functions, especially memory and learning, as well as executive and motor functions, complex attention, social cognition, and language [1]. The estimated proportion of the general population with dementia is around 50 million of people worldwide, and 60% of those cases correspond to AD [2]. It begins with a symptomatic stage of cognitive decline, called Mild Cognitive Impairment (MCI), characterized by an impairment in cognition that is not severe enough to compromise social and/or occupational functioning [3]. As the progression of this disease lasts for decades, from the appearance of the first sign to the onset of severe clinical symptoms, the clinician's first challenge is to identify the first significant cognitive changes [4]. Indeed, the diagnosis of dementia is usually made when the patient is at least partially dependent on his/her family members [5]. However, a timely diagnosis can facilitate care and support patients and their families in managing this very disabling disease [6].

Conventional techniques to detect AD are costly and distressing. However, electroencephalography (EEG) is a fast, inexpensive, and noninvasive technique to gather brain data, but its interpretation requires a visual inspection, which is often time-consuming and varies with the expertise experience. Moreover, when the EEG recording is long, then its manual

review requires a lot of time with the risk of errors because of the presence of artifacts in the signal. Thus, automated methods based on EEG signal analysis in combination with supervised machine learning have become an important topic of research to assist clinicians in the challenging task of early AD detection [7].

The technique for the sampling of EEG signal consists of placing electrodes on the scalp according to a certain configuration, and the most commonly used is the international 10–20 system [8]. The electrodes record the postsynaptic biopotentials of all the neurons with the same spatial orientation in order to map the electrical activity of the cerebral cortex. The biopotentials, which are sampled in bipolar mode by different electrode pairs or in monopolar mode with a reference electrode, constitute the raw signals. Subsequently, preprocessing procedures clean the raw signals from artefacts and apply band-pass filtering to reject out-of-band noise. Basically, all preprocessing steps convert the raw signals into EEG signals [9]. The composition of the EEG signal is complex, but it can be divided into five frequency bands: delta (1–4 Hz, $\delta$), theta (4–8 Hz, $\theta$), alpha (8–13 Hz, $\alpha$), beta (13–30 Hz, $\beta$), and gamma (30–40 Hz, $\gamma$) [10].

Features are extracted from EEG data through a procedure denominated *feature extraction*. Features should be independent and discriminative to facilitate the classification of the subjects. Usually, features such as complexity, coherence, or spectral power are extracted from the time and frequency domains [7]. Fourier transform (FT) is the main technique used to extract frequency domain features for AD detection. Nonetheless, EEG signals are nonstationary and nonlinear in nature. The wavelet transform (WT), which decomposes a signal into the combination of functions (wavelets) of finite length and different frequencies, represents a suitable alternative to address this issue [11]. On the contrary, the frequency domain represents the principal source of the EEG features for AD detection. Indeed, different changes in the frequency patterns of the brain waves have been found in MCI and AD patients compared to healthy aged subjects [12]. All feature extraction methods extrapolate EEG signal features from different domains (e.g., frequency and time) [13]. Then, statistical or machine learning analyses [14] can use these features to develop and validate models based on linear or nonlinear systems [15] to distinguish AD from MCI or normal aging. In particular, supervised machine learning (SML) permits the development of robust classification models for recognizing AD [16], frontotemporal dementia [17], and other pathologies [18]. Complex and heterogeneous symptoms complicate the diagnosis, because more often than not, biomarkers are intrinsically hidden in the EEG signal.

Rhythms are often used to analyze the EEG in a particular sub-band through filtering, due to the different activities between the frequency bands [19,20]. In fact, previous studies have shown that the relative power in fast rhythms ($\alpha$ and $\beta$) decreases, while, in slow rhythms ($\delta$ and $\theta$), it increases [21,22]. This effect shifts the peak power towards lower frequencies, which is why it is also called "shift-to-the-left" (STTL) [23]. The method presented in this paper exploits the power intensity of EEG signals, filtered in the time domain by using both high-pass and a low-pass filters in order to analyze the STTL phenomenon. Our idea is to classify the absolute difference in power between fast and slow rhythms for each individual, using it as a biomarker. For this purpose, we also use the power spectrum density (PSD) calculated with the help of the spectrogram and SML to choose the best filter cutoffs and improve the classification performance.

## 2. Materials and Methods

We used an EEG dataset composed of 109 EEG recordings (49 AD, 37 MCI, and 23 HC) collected in resting condition and with closed eyes at the IRCCS Centro Neurolesi "Bonino-Pulejo" in Messina (Italy). A diagnosis of AD or MCI was formulated following the guidelines of the Diagnostic and Statistical Manual of Mental Disorders (fifth edition, DSM-5).

### 2.1. Data Acquisition and Preprocessing

Multi-channel EEG signals were recorded by using 19 electrodes placed according to the 10–20 system [8] in monopolar connection with the earlobe electrode as a reference.

Raw electrical brain activity (μV) recordings last about 300 s. For more details on data collection, the reader can refer to the previous study [24].

In the preprocessing step, the sampling rate is normalized to 256 Hz, and EEG are filtered at a 1-Hz low cutoff (high-pass) and at a 30-Hz high cutoff (low-pass). After filtering, artifacts are detected by visual inspection and rejected. One hundred and fifty seconds of cleaned EEG are considered for each subject, extracted from the central part of the EEG signal in order to maintain the maximum signal information to train classifiers on the same length signals but without losing too many instances. Thus, four subjects were excluded (i.e., 1 AD and 3 HC) due to an excessive number of artifacts, and the dataset dropped to 105 EEG recordings.

### 2.2. Feature Extraction

The feature extraction procedure includes two main steps: (i) data exploration in the time–frequency domains by means of PSD computed in the spectrogram and (ii) construction of the double digital filter and its application.

### 2.2.1. Data Exploration in the Time–Frequency Domains

For each subject, we generated a unique signal by concatenating the 19 biopotentials signals (i.e., one for each electrode). This concatenated signal provides a complete view of the whole subject's signal and allows to know the electrodes more involved in the STTL phenomenon. Therefore, the 3 classes (AD, MCI, and HC) contain as many concatenated signals as there are subjects in each of them. Then, the average of each class is calculated, resulting in 3 average signals. Therefore, we apply the MATLAB *pspectrum* function (it is included in the Signal Processing Toolbox introduced in version R2017b), setting the spectrogram mode and providing input in the sampling frequency ($f_s$) and the signal in the time domain. The power spectrum density of the signal is computed, also performing the Short-Time Fourier Transform (STFT) of the signal and evaluates its power [25,26]. This step allows to find the best cutoff frequencies ($f_{cut}$) that can separate the classes. For the sake of clarity, recall that the spectrogram is a function used to plot the STFT of the signal, determining both the sinusoidal frequency and phase contained in different time frames and composing the entire signal.

### 2.2.2. Double Digital Filter Construction

The second step includes the construction of two Finite Impulse Response (FIR) digital filters, i.e., a second-order Butterworth filter for high-pass (FIR-H) and low-pass (FIR-L) frequencies by using the cutoff frequencies ($f_{cut}$) previously identified [27]. Thus, each EEG signals provided by an electrode is double-filtered, and two signals are generated in the time domain, called $EEG^{(L)}$ and $EEG^{(H)}$, where the first value is the EEG filtered with FIR-L, while the second value is the EEG filtered with FIR-H. Subsequently, we compute the power (see Appendix A) of these two signals, $P_{xx}^{(L)}$ and $P_{xx}^{(H)}$, with the aim to calculate the square of their absolute difference:

$$P_{(L-H)}^2 = \left\| P_{xx}^{(L)} - P_{xx}^{(H)} \right\|^2 \tag{1}$$

This value is the extracted feature corresponding to our biomarker. Figure 1 shows a schematic representation of the double digital filter construction.

In this way, we are able to represent each initial EEG signal as an array of values sampled in a single feature, $P_{(L-H)}^2$, reducing the input size. This procedure is iterated for all subjects, as shown in Table 1. To ensure that the cutoff frequencies identified in the first stage of the procedure are really the best for class separation, we vary $f_{cut}$ from 1 Hz to 18 Hz with a step size of 1 Hz, in order to achieve the best class separation.

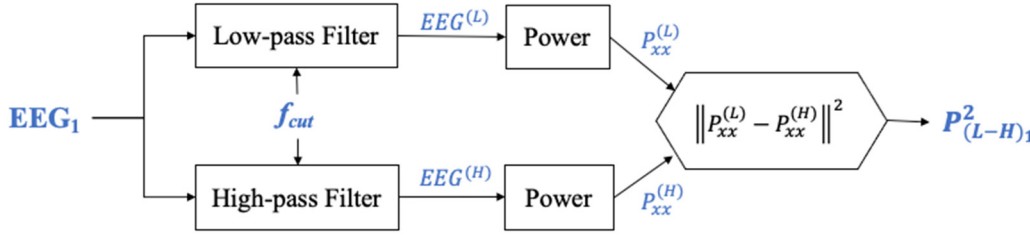

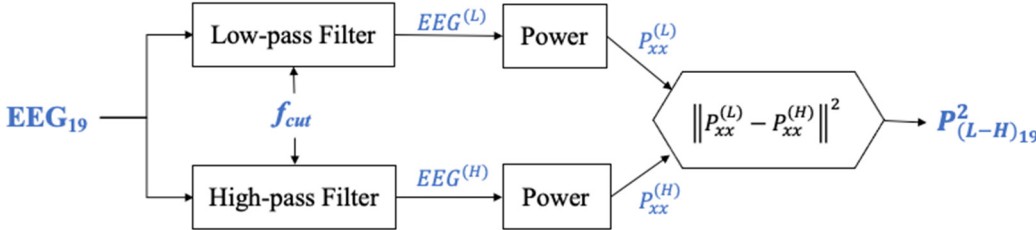

**Figure 1.** Schematic representation of the double filtering method. The initial input is the subsequence of the main EEG signal, which is divided and filtered according to the two main branches. The final result is the square of the power absolute difference. The block "Power" is explained in Appendix A.

**Table 1.** Schema of the feature extraction procedure: features extracted from an EEG recording are shown in the last column.

| N-Subjects | EEG Signals | Label | Extracted Features |
|:---:|:---:|:---:|:---:|
| 1 | $EEG_1, EEG_2, \ldots, EEG_{19}$ | AD | $P^2_{(L-H)_1}, P^2_{(L-H)_2}, \ldots, P^2_{(L-H)_{19}}$ |
| 2 | $EEG_1, EEG_2, \ldots, EEG_{19}$ | MCI | $P^2_{(L-H)_1}, P^2_{(L-H)_2}, \ldots, P^2_{(L-H)_{19}}$ |
| $\ldots$ | $\ldots$ .. | $\ldots$ | $\ldots$ .. |
| 109 | $EEG_1, EEG_2, \ldots, EEG_{19}$ | HC | $P^2_{(L-H)_1}, P^2_{(L-H)_2}, \ldots, P^2_{(L-H)_{19}}$ |

The column Label shows the initial labeling used in supervised machine learning.

### 2.3. Classification

The signal classification is performed by an SML analysis through three classification methods: decision trees (DT), support vector machines (SVM), and k-nearest neighbor (KNN) [28]. The algorithms are implemented in Python (version 3.7.21) by means of the scikit-learn toolkit [29]. Python is an open-source programming language, and its choice stems from the availability of many external libraries, frameworks, and tools from a huge community distributed all over the world.

In this study, we define five classification problems: (i) AD vs. HC, (ii) AD vs. MCI, (iii) MCI vs. HC, (iv) AD + MCI vs. HC, and (v) AD vs. MCI vs. HC. The first four problems were addressed in our previous study [24], whereas the last one defines the three class classification problem [30]. Then, for each problem, we perform 10 runs where the extracted features are randomly sampled. Finally, we adopt the following procedure:

1. Dataset splitting in training 70% and data tests 30%, except for the (v) case where the data has been divided into 80% training and 20% data tests;
2. Dataset size reduction with the Linear Discriminant Analysis (LDA) [31];
3. Application of the three aforementioned supervised machine learning methods;
4. Tuning of the hyperparameters of the machine learning algorithms combined with k-fold cross validation [32];
5. Data validation and performance evaluation through the confusion matrices.

Regarding point 1, data splitting can affect the performance of evaluators, so making appropriate decisions during this step is very challenging, as highlighted in [33]. Here, the authors summarize the challenges of data splitting into three main points: (i) Data imbalance, (ii) Data loss, and (iii) Concept drift. Taking these points into consideration, we divide the initial datasets into 70–30 and 80–20. The first choice shows very high levels of accuracy in the first four cases of the classification; on the contrary, 80–20 has an accuracy increase of 5% in the three-class classification problem. In order to reduce the less meaningful features, we applied the LDA to the extracted features. The LDA, cited at point 2, is a well-known data mining algorithm [34] and is suitable in those cases where classes are unbalanced or nonlinearly separable, such as the EEG signal. LDA automatically defines a separation hyperplane between the points that belongs to a class by generating two subclasses from the main one. Consequently, the Fisher criterion [35], also called Fisher's linear discriminant, maximizes the ratio of the between-class variance to the within-class variance in any particular dataset, ensuring the maximum separation. In this way, it is possible to discard those values of the extracted features that do not affect the variance of the main class, decreasing the sample dimensions.

With the purpose of improving the performance of the classification algorithms, we automatically introduce additional parameters, namely hyperparameters, provided by an external constructor. However, a wrong choice of the hyperparameters can lead to incorrect results and to obtaining a poor performance model [36,37]. In this work, we chose the grid search algorithm (GScv) [38], implemented through the python function GridSearchCV. GScv is the simplest algorithm for hyperparameter optimization [39]. However, it is time-consuming, since it considers all combinations to find the best point (Figure 2a), and each grid point needs cross-validation in the training phase.

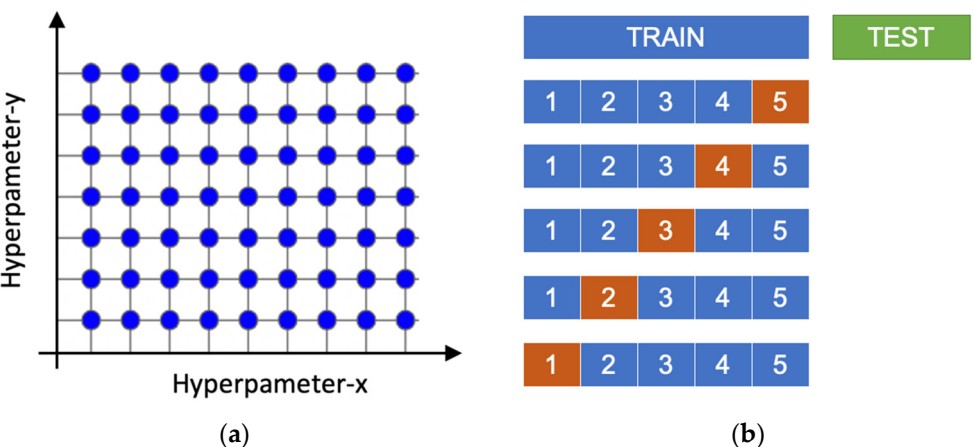

(**a**)            (**b**)

**Figure 2.** (**a**) Example of a grid search domain, where the hyperparameters are distributed into a matrix. (**b**) A schematic representation of 5-fold cross-validation. In (**b**), the data train (the blue boxes) is split in "k" subsegments, and one (the orange box) of them is used as the validation in each iteration.

In addition, the cross-validation procedure resamples the data randomly for the better evaluation of machine learning models. To improve the model validation, the procedure of cross-validation is iterated k times. Consequently, the data training is segmented into k subgroups [40]. For this reason, the procedure is often called k-fold cross-validation. Here, we split the training datasets into k = 10-fold without reinsertion, where 9-folds are used to train the model and 1-fold for the performance evaluation [41], as represented in Figure 2b. The estimator evaluation (e.g., accuracy) is the average of each estimator computed over the kth iteration.

### 3. Results

In this section, we report the results obtained after any steps of our method, e.g., from the feature extraction to the classification process, providing the accuracy measures of the classification algorithms.

First, we show in Figure 3 the allocation of the power spectrum for each patient. Looking closely at Figure 3, we can enhance that the PSD in AD and MCI is restricted in low frequencies (<7 Hz), while, in HC, the power is spread up to about 14 Hz. Thus, we expect that the next part of the feature extraction procedure also identifies that these frequencies are the best cutoffs for a good separation between classes.

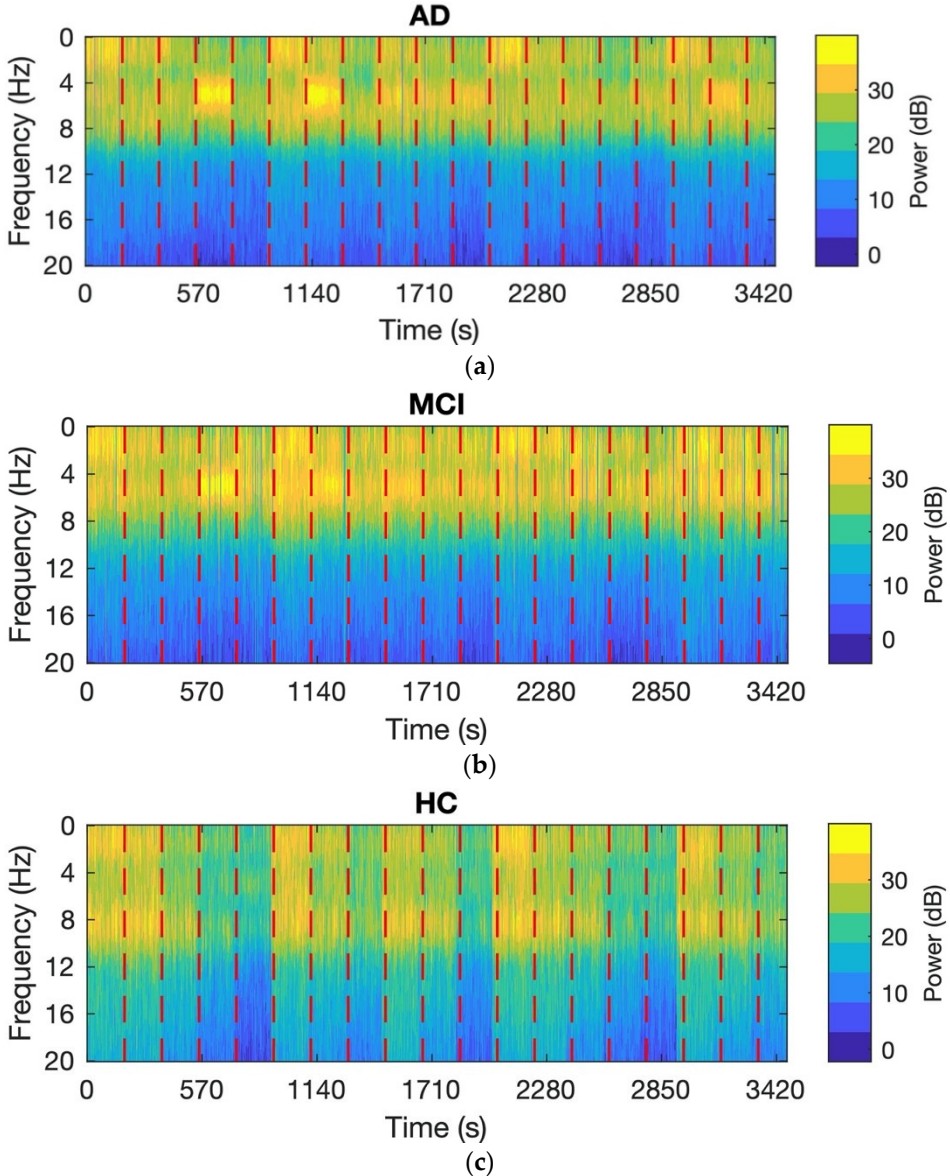

**Figure 3.** Computed pspectrum from 19 biopotentials (EEG signals) in a frequency range of 0–20 Hz for AD (**a**), MCI (**b**), and HC (**c**). The dashed red line separates each individual signal from the 19 biopotentials that make up the preprocessed EEG signal.

As viewable in Figure 4a, the statistical test carried out on the characteristics shows a good separation between classes in these $f_{cut}$ values. Indeed, we found an excellent class separation between HC and AD-MCI for the value of $f_{cut}$ equal to 7 Hz, as can be seen in Figure 4b, and an excellent separation between AD and MCI with $f_{cut} = 16$ Hz (Figure 4c).

These interesting results suggest applying the first filtering at 7 Hz to exclude controls and the subsequent filtering at 16 Hz for a better classification between MDI and AD.

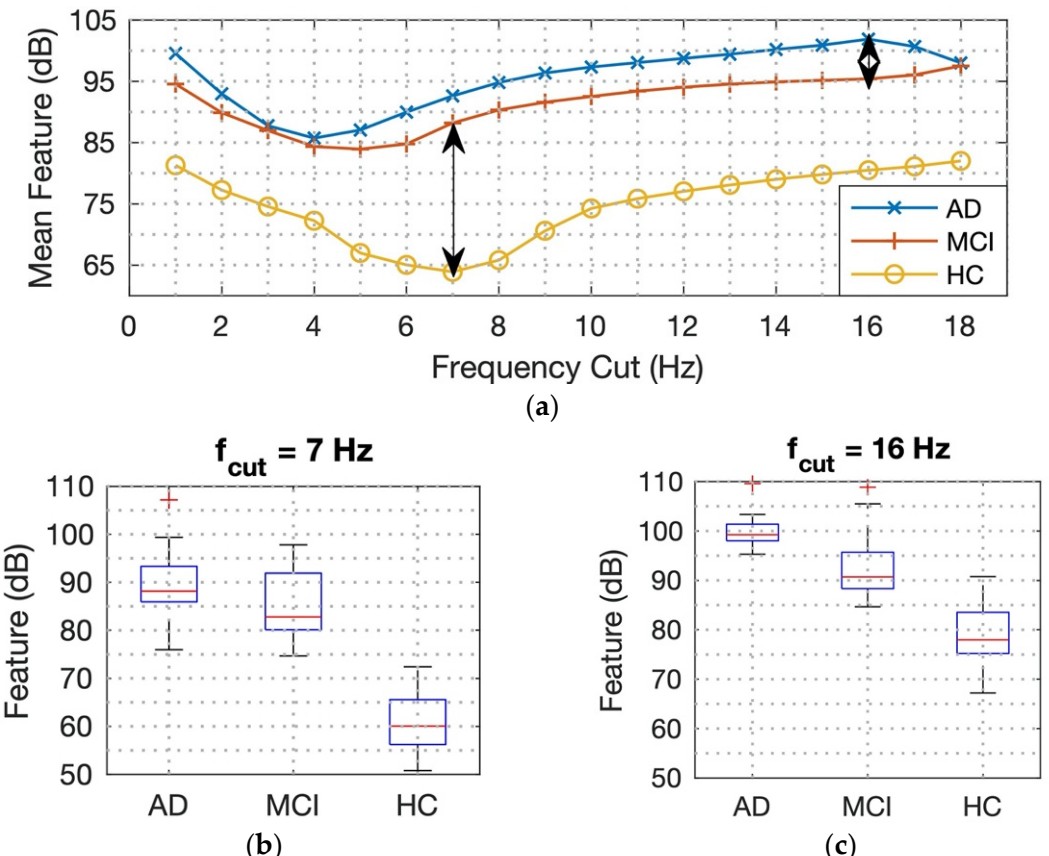

**Figure 4.** (**a**) The mean value of the feature extraction, the black double arrows, show the maximum distance between HC and AD + MCI for $f_{cut}$ = 7 Hz and between AD and MCI for $f_{cut}$ = 16 Hz. The statistical results are shown in (**b**) for 7 Hz and (**c**) for 16 Hz. In both of the last two graphs, the main box represents the data distribution, while the red line represents the median value, and the whisker stretches from the box show the range of the data, except for the outlays that are represented, such as the floating point (+).

The classification of the samples achieves a high level of accuracy, especially in distinguishing AD vs. HC and MDI vs. HC cases. In fact, the three classification procedures achieve an accuracy value of more than 87% and up to 97%, as shown in Figure 5a,b. Moreover, when we consider AD + MCI vs. HC, the accuracy reaches a value between 84% and 89% (Figure 5c). The effectiveness of the proposed method consists of tunneling $f_{cut}$, as shown in Figure 5d for the AD vs. MCI case. Here, the accuracy of the classification methods improves from 49–60% to 80–83% when $f_{cut}$ is increased from 7 Hz to 16 Hz. Finally, Figure 5e shows the comparison between the three different cases and reinforces the hypothesis of the effectiveness of the proposed method for feature extraction. In the latter case, the accuracy value is between 73% and 86%.

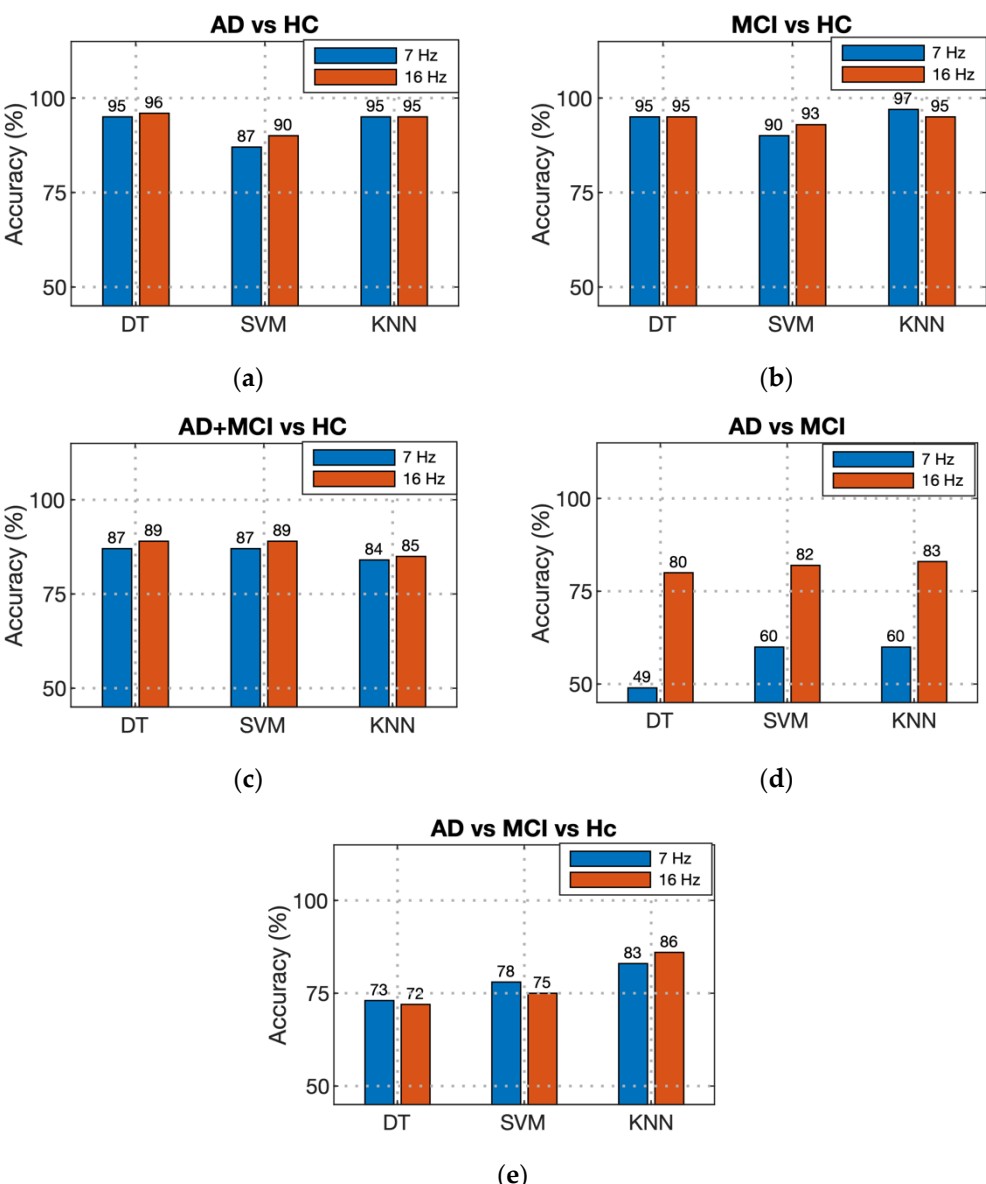

**Figure 5.** Each subfigure displays the comparison between the three classification methods: decision tree (DT), support vector machine (SVM), K-nearest neighbors (KNN) in the different problems: (**a**) AD vs. HC, (**b**) AD vs. MCI, (**c**) MCI vs. HC, (**d**) AD + MCI vs. HC, and (**e**) AD vs. MCI vs. HC. Furthermore, all the subfigures show the accuracy results achieved in percentages at $f_{cut}$ = 7 Hz in the blue column and $f_{cut}$ = 16 Hz in the red column.

In Table 2, we present the computation times of the classification procedure. The costs of feature extraction are low, and the computation time is about 0.1 s for a single subject. On the contrary, in the classification process, the computation time is higher because of the search for the best hyperparameters improving the accuracy. In the binary classification problems, the total execution time is, on average, 39.5 s, of which the DT takes about 21 s, the SVM takes about 15 s, and the K-NN takes about 3 s, except in the last case, when the execution time increases by two seconds. K-NN is the algorithm with the lowest runtime, while DT is the slowest. This is because KNN stores the training data in an n-dimensional space defining the pattern's spaces, and for each unknown sample, it assigns the pattern's space with a minor distance function [42] while DT extracts a classification model composed of features and value assignments requiring a longer runtime, although DT provides a better interpretation of the classification solution.

**Table 2.** Average execution time of the classification algorithm, including tuning of the hyperparameters.

| Case | $f_{cut}$ (Hz) | Time (s) DT/SVM/K-NN | Tot. Time (s) |
| --- | --- | --- | --- |
| AD vs. HC | 7 | 21.8/14.0/3.0 | 38.8 |
| | 16 | 20.9/14.0/3.0 | 38.0 |
| AD vs. MCI | 7 | 22.3/14.8/3.0 | 40.1 |
| | 16 | 21.1/15.0/3.0 | 39.1 |
| MCI vs. HC | 7 | 20.7/13.3/3.0 | 37.0 |
| | 16 | 21.1/13.6/3.0 | 37.7 |
| AD + MCI vs. HC | 7 | 20.6/16.4/3.0 | 40.0 |
| | 16 | 21.1/18.5/3.0 | 42.6 |
| AD vs. MCI vs. HC | 7 | 25.0/22.6/4.0 | 51.6 |
| | 16 | 25.9/24.1/4.8 | 54.8 |

The Time column shows the execution time of each classifier, where DT = decision tree, SVM = support vector machine, and K-NN = K-nearest neighbors. The last column shows the sum of the execution times.

## 4. Discussion

In this paper, we proposed a new method for the feature extraction in AD recognition from EEG signals. Our findings confirm that AD affects the power spectrum of the patient, according to previous studies [43,44]. The proposed method is carried out in the time domain, exploiting the knowledge a priori of the EEG signals, i.e., power spectrum, spectral entropy, and phase synchronization. We used three different classification algorithms to validate our method, obtaining promising results. Indeed, the accuracy ranged between 73% and 97%, overcoming previous studies [24,45]. In particular, in [45], the best accuracy rate was 94%, obtained by using discrete wavelet transform to extract features, whereas, in [24], we used a DT classifier (i.e., the C4.5 algorithm), reaching the following levels of accuracy: 83%, 92%, 79%, and 73% in HC vs. AD, HC vs. MCI, MCI vs. AD, and HC vs. MCI + AD classification problems, respectively. Furthermore, in the last two cases, we increased the level of accuracy, with values higher than 80% combining in the information gain filter. Here, K-NN was the best classification algorithm as concerning the accuracy (almost always greater than 80%), as shown in Figure 5, with a runtime of about 3 s (Table 2). On the contrary, the accuracy of DT was the lowest, and its running times the highest because of the computation complexity in the construction of the solution in a human interpretable format.

The recent literature has reported several techniques using EEG signals for the early diagnosis of AD, differing in how features are extracted. Some of these, such as event-related potentials, signal a complexity analysis and relative power, involving a time domain signal analysis. Other techniques, instead, work in the Fourier domain, such as the coherence metric that evaluates the synchrony between two signals. In addition, there are techniques exploiting the analysis in the frequency domain, such as the continuous or discrete wavelet transform [7]. Cejnek et al. [46] employed a linear neural unit with gradient descent adaptation as the filter to predict AD, achieving a specificity of 85% and a sensitivity between 85% and 94%, depending on the classifier. In Reference [47], the authors developed an algorithm that consists of three cascade methods for analysis: discrete WT, PSD, and coherence. They tested this method on 35 subjects by means of the bagged trees classifier trained with five-fold cross-validation, obtaining a 96.5% accuracy.

There are only a limited number of works that have exploited time–frequency or bispectrum-based features, such as discriminating coefficients. The multimodal machine learning approach of Ieracitano et al. [48], where EEG signals are projected into the time–frequency domains by means of the continuous WT to extract a set of features from EEG sub-bands, while the nonlinear the phase-coupling information of EEG signals is also used to generate a second set of features from the bispectrum representation. This method provides high levels of accuracy with different classifiers in all considered problems: AD

vs. HC, Ad vs. MCI, MCI vs. HC, and AD vs. MCI vs. HC and on a large cohort of subjects (i.e., 189 subjects: 63 AD, 63 MCI, and 63 HC). Similar to our method for the early detection of AD, the Lacosogram tool [21] performs a statistical analysis to measure distances between EEG sub-bands, obtaining an accuracy of 98.06% for HC vs. MCI, 95.99% for HC vs. AD, and 93.85% for MCI vs. AD. Kulkarni and Bairagi in [49] decomposed EEG by using the WT to decompose the EEG signal into its five sub-bands. The means and variances of the wavelet coefficients were evaluated and used as input to a SVM classifier, achieving an accuracy value of 88% in AD vs. HC classification.

Although, in our previous work [24], the experimental results showed that wavelet coefficients evaluated by applying the discrete wavelet transform achieved the highest accuracy rates (i.e., 83.3% for AD vs. HC, 91.7% for MCI vs. HC, and 79.1% for AD vs. MCI), the three-class classification did not achieve good results when we used only Fourier or Wavelet transform. On the contrary, here, the results of the three-class classification problem (AD vs. MCI vs. HC) were also reported, showing an average accuracy of 78% for the three classifiers and longer running times. This further proved the validity of this feature extraction method, which plays a key role in the analysis. Indeed, the method separates the high frequencies and the low frequencies of the EEG signal, and then, it computes their powers. The comparison of these powers shows an imbalance of energies in the frequency range, demonstrating the phenomenon of the STTL described in [15].

Our findings showed that a correct choice of $f_{cut}$ increases the accuracy of discrimination between AD, MCI, and HC subjects. In addition, the proposed method for feature extraction is simple and fast in running time, and therefore, it is easily replicable in different development environments. This is undoubtedly its greatest strength, making its implementation and understanding very easy. The model, tested on a larger sample, could lead to the identification of biomarkers capable of determining features that discriminate electrical signals between different AD cohorts at specific electrodes.

As a future work, we plan to improve the classification method to the point of removing all the HC subjects, applying double filtering with $f_{cut}$ = 7 Hz from the main sample, and distinguish AD from MCI by applying double filtering with $f_{cut}$ = 16 Hz. However, given the heterogeneity of the disease, a larger cohort is necessary to confirm the results of this study. We also plan to test the method in different EEG recording protocols, maybe while the subject performs a cognitive task, in order to provide insights on how AD affects certain cognitive areas.

Since AD is expected to affect a large part of the worldwide population in the following years, EEG represents a suitable technique to assist clinicians in an early diagnosis. From this perspective, the method could be implemented on embedded devices and used in real time during EEG signal acquisition due to its low computational resource requirements. Considering the simplicity and robustness of the double filtering, we could promote it as an inexpensive and portable software suite by programming the current embedded electronic microprocessor, such as a Dev Board [50].

**Author Contributions:** Conceptualization, E.W. and D.P.; methodology, D.P.; software, D.P. and P.D.P.; validation, E.W., M.C.D.C. and S.D.S.; formal analysis, E.W. and D.P.; investigation, M.C.D.C.; resources, M.C.D.C.; data curation, S.D.S.; writing—original draft preparation, D.P., M.C.D.C. and E.W.; writing—review and editing, E.W. and M.C.D.C.; supervision, E.W. and M.C.D.C.; project administration, E.W. and M.C.D.C.; and funding acquisition, M.C.D.C. All authors have read and agreed to the published version of the manuscript.

**Funding:** This study was supported by Current Research Funds 2021, Ministry of Health, Italy.

**Institutional Review Board Statement:** The Ethical Committee of the IRCCS Centro Neurolesi "Bonino-Pulejo" approved the study after informed consent to participate in the study was signed by the enrolled subjects (reference number 40/2013).

**Informed Consent Statement:** Informed consent was obtained from all subjects involved in the study.

**Data Availability Statement:** The datasets analyzed during the current study are available upon request from the IRCCS Centro Neurolesi "Bonino-Pulejo" (M.C.D.C.).

**Conflicts of Interest:** The authors declare no conflict of interest.

## Appendix A

In this section, we explain how we calculate the power of the EEG signal after splitting the signal into the high- and low-frequency components. This operation describes the function of the block "Power" in Figure 1. The power of a generical signal $s(t)$ in a time interval or period $T$ is calculated as [51]:

$$P_{xx} = \frac{1}{T} * \int_{t_0}^{t_0+T} \|s(t)\|^2 \, dt \tag{A1}$$

In our work, we considered $t_0 = 0$, and the EEG is a sampled signal, i.e., a discrete-time signal, so that (A1) can be rewritten as:

$$P_{xx} = \frac{1}{N} * \sum_{n=0}^{N} EEG(n)^2 \tag{A2}$$

where $n$ is the index of the $n$th sample that compose the EEG, and $N$ is the total number of samples. As well-known, the ratio between $N$ and the sampling rate returns $T$, and this indicates the linear dependence between $N$ and $T$.

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
