# Peer review of "EEG Signal Processing and Supervised Machine Learning to Early Diagnose Alzheimer’s Disease"

_applsci, doi:10.3390/app12115413_

Round 1

Reviewer 1 Report

I think this article's idea and the experimental process are good, but many signs indicate it was finished in a hurry and without detailed examination.

1. Problems with the English writing (the following only lists a few. In fact, many sentences are problematic or very difficult to read):

a) Some English expressions need to be improved. It is recommended to refer more to the expressions in academic peer articles. E.g., EEG signal"s" analysis.

b) There are many errors in the grammar of some sentences, e.g.:
    (114) "The initial input is the sub-sequence of the main EEG signal, which are splited and filtering according the two main branches." Four problems appear here: "are", "splited", "filtering", and "according".
    (130) "2th" order Butterworth filter
    (169) "on the contrary, the 80-20 has an accuracy of more than 5% in the three-class classification problem." -> has an accuracy increase of... ("more than 5%" is not an accuracy, but an accuracy increment)
    (280) Indeed, the method separate"s" the high frequencies and the low frequencies of the EEG signal, and then it compute"d" their powers.

c) Some minor typos need to be modified, e.g.: (109/110) "i)" "(ii)"

2. Problems with the format:

There is no need to use asterisks in Table 1. Just write the contents of the asterisk directly in the caption. Similar to Table 2.

The figure after (234) has no caption.

(263) "It WORKS IN THE TIME DOMAIN exploiting the power spectrum, spectral entropy, and phase synchronization." Consider a precise expression.

Table 2: "classifiers" and "time" can be combined. It does not make sense to list a metric that is the same for every row.

Author Response

I think this article's idea and the experimental process are good, but many signs indicate it was finished in a hurry and without detailed examination.

We would like to thank the reviewer for his appreciation and constructive criticism. We followed all his/her suggestions during the revisions, reading with attention the manuscript looking for errors and mistakes.

  1. Problems with the English writing (the following only lists a few. In fact, many sentences are problematic or very difficult to read):

  1. a) Some English expressions need to be improved. It is recommended to refer more to the expressions in academic peer articles. E.g., EEG signal"s" analysis.

 Done

  1. b) There are many errors in the grammar of some sentences, e.g.:

    (114) "The initial input is the sub-sequence of the main EEG signal, which are splited and filtering according the two main branches." Four problems appear here: "are", "splited", "filtering", and "according".

    (130) "2th" order Butterworth filter

    (169) "on the contrary, the 80-20 has an accuracy of more than 5% in the three-class classification problem." -> has an accuracy increase of... ("more than 5%" is not an accuracy, but an accuracy increment)

    (280) Indeed, the method separate"s" the high frequencies and the low frequencies of the EEG signal, and then it compute"d" their powers.

Done

  1. c) Some minor typos need to be modified, e.g.: (109/110) "i)" "(ii)"

Done

  1. Problems with the format:

There is no need to use asterisks in Table 1. Just write the contents of the asterisk directly in the caption. Similar to Table 2.

Done

The figure after (234) has no caption.

We added the caption to Figure 4.

(263) "It WORKS IN THE TIME DOMAIN exploiting the power spectrum, spectral entropy, and phase synchronization." Consider a precise expression.

Done

Table 2: "classifiers" and "time" can be combined. It does not make sense to list a metric that is the same for every row.

 Done

Reviewer 2 Report

The research and its results presented in this paper show promising possibility to support the early detection of AD and MCI with machine learning support. The preprocessing with feature extraction plays an important role in the described and analyzed method.

I miss some information about the cases which give the samples for the machine learning, it is hard to see whether the samples are from the early period of AD and  MCI.

The writing has basically good quality. Some minor problems/questions you can find below.

many times: EEG signals analysis - EEG signal analysis is more frequently used

line 115: splited -> splitted ; filtering -> filtered ; according the -> according to the

line 142: features extraction -> feature extraction

line 146: a SML -> an SML

line 147: Usage of capitals in algorithm names is not consistent.

line 166: authors summaries -> authors summarize

line 193: Example of a grid search domain the x-axis and y-axis have the same label - bad sentence

lines 194-195: The data train are splited in k=5 segment and one of the is used as a validation in each iteration. - bad sentence

The figure 4 does not have a caption.

Its placement is left compared to fig 3 and fig 5.

line 239: consists in -> consists of

The comparison of the performance three algorithms would give more information with better understanding the parameter behavior. Maybe a little weaker precision can be achieved with much less performance need, so the results using different parameters could also be interesting.

line 252: In fact, K-NN only performs a comparison based on distance... - what does it mean?

lines 274-275: Can we state that the lower precision and higher running time is because the model built is human interpretable?

lines 280-282: Present and past tense mixed improperly.

Author Response

The research and its results presented in this paper show promising possibility to support the early detection of AD and MCI with machine learning support. The preprocessing with feature extraction plays an important role in the described and analyzed method.

We agree with the reviewer and would like to thank him/her for his appreciation and constructive criticism.

I miss some information about the cases which give the samples for the machine learning, it is hard to see whether the samples are from the early period of AD and MCI.

The patients were classified either in AD or MCI, taking into account the World Health Organization standard. Moreover, we added a subsection reporting data information and its pre-processing.

The writing has basically good quality. Some minor problems/questions you can find below.

We read with attention the manuscript looking for errors and mistakes, besides to correct their suggested by the reviewer.

many times: EEG signals analysis - EEG signal analysis is more frequently used

Done

line 115: splited -> splitted ; filtering -> filtered ; according the -> according to the

Done

line 142: features extraction -> feature extraction

Done

line 146: a SML -> an SML

Done

line 147: Usage of capitals in algorithm names is not consistent.

Done

line 166: authors summaries -> authors summarize

Done

line 193: Example of a grid search domain the x-axis and y-axis have the same label - bad sentence

We have rewritten this sentence.

lines 194-195: The data train are splited in k=5 segment and one of the is used as a validation in each iteration. - bad sentence

The figure 4 does not have a caption.

We have inserted the caption

Its placement is left compared to fig 3 and fig 5.

Done

line 239: consists in -> consists of

Done

The comparison of the performance three algorithms would give more information with better understanding the parameter behavior. Maybe a little weaker precision can be achieved with much less performance need, so the results using different parameters could also be interesting.

The reviewer has raised a very interesting point, in this paper we emphasize the simplicity and robustness of the proposed feature extraction method. However, it is possible to understand which channels provide more information by looking at the spectrogram plot. As a result, we can exclude the less relevant channels from the study.

line 252: In fact, K-NN only performs a comparison based on distance... - what does it mean?

We have rewritten the sentence, exploiting that the K-NN stores all the data-training without using a model defined a-priori, but it assigns the new sample to the pattern space closest via a distance function.

lines 274-275: Can we state that the lower precision and higher running time is because the model built is human interpretable?

We have rewritten this sentence.

lines 280-282: Present and past tense mixed improperly.

Done

Reviewer 3 Report

This work describes a method to discriminate among patients affected by Alzheimer's disease (AD), Mild Cognitive Impairment (MCI) and 16 healthy controls (HC) by analysing EEG signals. In this method, signal processing and machine learning techniques were applied. This study look interesting, but the authors need to address below a few points to improve the clarity and quality of this work.

1) The authors claimed that 180 out of 300 seconds were used for analysis. It is not clear to readers which part of the signals were retained and why 40% of the signal were thrown away.

2) It is not clear to readers how and why the authors concatenated the signals within each patient, especially given that the authors listed 19 features per patient in Table 1

3) The authors need to give details about parameter setting in spectrogram

4) In Figure 3, the maximum frequency shown is  14 Hz, but later the authors used 16 Hz as a cutoff frequency. The authors are encouraged to include high frequency components in Figure 3.

5) The authors only draw performance comparison between 7 and 16 Hz as cutoff frequency. What about other options (8-15 Hz)? 

6) The methods section described grid search, but the only parameter tuned was cutoff frequency; there was not grid. The authors may consider shortening the description on that. 

7) It was not clear to readers how the authors applied LDA, as they described in the methods.

8) Model performance was compared and reported on validation sets that tuned hyperparameters. The authors did not seem to use a hold-out set to report the final result.

9) No comparison with state-of-the-art.

Author Response

This work describes a method to discriminate among patients affected by Alzheimer's disease (AD), Mild Cognitive Impairment (MCI) and 16 healthy controls (HC) by analysing EEG signals. In this method, signal processing and machine learning techniques were applied. This study look interesting, but the authors need to address below a few points to improve the clarity and quality of this work.

We would like to thank the reviewer for his appreciation and constructive criticism. We followed all his/her suggestions during the revisions in order to improve the manuscript.

1) The authors claimed that 180 out of 300 seconds were used for analysis. It is not clear to readers which part of the signals were retained and why 40% of the signal were thrown away.

We thank the reviewer for the helpful critique, which also gave us an opportunity to note another error: the signals used were 150sec long! The idea was of running the method on same length signals. In addition, the literature in this field reports the use of EEG recording of at least 120sec. 

2) It is not clear to readers how and why the authors concatenated the signals within each patient, especially given that the authors listed 19 features per patient in Table 1

The concatenation of such 19 signals produces a unique signal on which we apply the pspectrum, in order to know the electrodes more involved in the "shift-to-the-left" phenomenon. This single concatenated signal (instead of seeing 19 signals) gives us a complete view through a single graph. We explained that in the manuscript.

3) The authors need to give details about parameter setting in spectrogram

We thank the Reviewer for his/her advice. First, we want to clarify that it is not a spectrogram but the “pspectrum function” used in spectrogram mode. This function computes and graphics the power spectrum in the frequency-time domain, taking into input the signal in the time domain and the sampling frequency, both of them are known during the signal acquisition. We added the input parameter used.

4) In Figure 3, the maximum frequency shown is 14 Hz, but later the authors used 16 Hz as a cutoff frequency. The authors are encouraged to include high frequency components in Figure 3.

We extend the frequencies till to 20 Hz.

5) The authors only draw performance comparison between 7 and 16 Hz as cutoff frequency. What about other options (8-15 Hz)?

We have extended the performance till to 18 Hz and we tested different cut-off values: the better separation among classes remains when we use 7 and 16Hz. We added a figure to show it (Figure 4(a))

6) The methods section described grid search, but the only parameter tuned was cutoff frequency; there was not grid. The authors may consider shortening the description on that.

We have shortened this description. 

7) It was not clear to readers how the authors applied LDA, as they described in the methods.

We have better described it.

8) Model performance was compared and reported on validation sets that tuned hyperparameters. The authors did not seem to use a hold-out set to report the final result.

We described the hold-out set when we explain that we have used the splitting 70-30% for all case except in three case comparison when we used the 80-20%. 

9) No comparison with state-of-the-art.

We did it within the discussion section.

Round 2

Reviewer 2 Report

The paper was corrected according to the suggestions of the reviewers, there are, however, few minor mistakes in the document, see examples below.

line 123: contains -> contain

line 125: . , - typo

line 126: R2017b) - unnecessary ")"

lines 285-286: Capital usage is inconsistent.

Author Response

The paper was corrected according to the suggestions of the reviewers, there are, however, few minor mistakes in the document, see examples below.

Thanks to the reviewers for their constructive criticism that allowed us to improve the manuscript. We read the manuscript again looking for errors and mistakes, besides to correct those suggested by the reviewer below.

line 123: contains -> contain

Done

line 125: . , - typo

Done

line 126: R2017b) - unnecessary ")"

It closes “(included…” at line 125

lines 285-286: Capital usage is inconsistent.

Done

Reviewer 3 Report

NA

Author Response

Thanks to the reviewer for his/her constructive criticism that allowed us to improve the manuscript.